# Evidence of an upper ionospheric electric field perturbation correlated with a gamma ray burst

Earth's atmosphere, whose ionization stability plays a fundamental role for the evolution and endurance of life, is exposed to the effect of cosmic explosions producing high energy Gamma-ray-bursts. Being able to abruptly increase the atmospheric ionization, they might deplete stratospheric ozone on a global scale. During the last decades, an average of more than one Gamma-ray-burst per day were recorded. Nevertheless, measurable effects on the ionosphere were rarely observed, in any case on its bottom-side (from about 60 km up to about 350 km of altitude). Here, we report evidence of an intense top-side (about 500 km) ionospheric perturbation induced by significant sudden ionospheric disturbance, and a large variation of the ionospheric electric field at 500 km, which are both correlated with the October 9, 2022 Gamma-ray-burst (GRB221009A).

Evidence of ionospheric disturbance induced by a gamma-ray burst (GRB) was first reported in 1988 by Fishman and Inan[1] as due to the GRB occurred on 1st August 1983, the strongest ever observed at that time, with a total fluence exceeding $10^{-3} ergs/cm^2/s$. The measured bulk effect on the ionosphere was the amplitude change of Very Low Frequency (VLF) radio signals, proof of the perturbation induced in the lower part of the ionosphere by that very energetic extrasolar event.

During cosmic GRB (and solar flare too), the intense high energy photon flux can abnormally ionize the lower ionosphere[2] by producing a large increase of free electron density[3]. As a consequence, the electron density grows giving rise to a variation of the ionospheric conductivity leading to a pronounced alteration in both VLF and ELF (Very Low and Extremely Low Frequency) electric field behaviour, respectively. Using ground VLF emitters, Inan et al.[4] showed that, if the burst is sufficiently severe (total fluence exceeding $10^{-3} ergs/cm^2$) and long-lasting, the ionospheric perturbation caused by a GRB can be observed in the bottom-side ionosphere (from about 60 km up to about 350 km of altitude). Although dedicated satellites recorded an average of more than one GRB per day in the last decade, intensive ionospheric reactions were seldom observed. In fact, only a handful of papers have reported the detection of ionospheric perturbations due to GRBs events[3–8], though always in the bottom-side ionosphere.

In addition, both Sentman et al.[9], and Price and Mushtak[10] have investigated GRB effects on Earth's ionosphere finding no significant variation on the ELF electromagnetic wave data. Nonetheless, Tanaka et al.[11] reported a clear detection of transient ELF signal caused by the December 27, 2004, event, a very intense cosmic gamma-ray flare, inducing a clear variation in the ionospheric Schumann resonance[12] detected by electromagnetic ground stations.

In this work we present the evidence of variation of the ionospheric electric field at about 500 km induced by the strong GRB occurred on October 9th, 2022. Using both satellite observations and a new ad hoc developed analytical model, we prove that the GRB221009A deeply impacted on the Earth's ionospheric conductivity, causing a strong perturbation not only in the bottom-side ionosphere[13,14], but also in the top-side ionosphere (at around 500 km).

## Results

On October 9th, 2022, at 13:21 UT, a highly bright and long-lasting GRB (hereafter GRB221009A), triggered many of the X and Gamma-ray space observatories, in particular Swift[15,16], Fermi[17,18], MAXI[19], AGILE[20,21] and INTEGRAL[22,23]. The GRB follow-up was observed by most operative telescopes in space and on-ground. The INTEGRAL (see Integral satellite data section for more details) gamma-ray observatory[24] detected the GRB both using the SPI spectrometer (SPectrometer of Integral) and the IBIS imager (Imager on-Board the INTEGRAL Satellite) as a complex, impulsive, very strong photon signal followed by a very intense gamma-ray afterglow[25]. The GRB221009A zenith was located

✉e-mail: mirko.piersanti@univaq.it

over India and the GRB photon flux was illuminating Europe, Africa, Asia and part of Australia (Fig. 1).

The light curves from the SPI detector and the IBIS imager (Fig. 2) show a multi-peaked structure with a moderately intense precursor, starting at 13:16:58 UTC, followed by a very strong prompt GRB emission, peaking at 13:21 UTC and, a long-lasting, sustained, soft gamma afterglow detected by both instruments in the energy range 75–1000 keV (SPI) and 0.250–2.6 MeV (IBIS), respectively. Optical follow-up with the OSIRIS (Optical System for Imaging and low-Intermediate-Resolution Integrated Spectroscopy) at the 10.4m GTC (Gran Telescopio CANARIAS) telescope confirmed the presence of a strong optical afterglow in the range 3700-10000Å with features suggesting a supernova progenitor[26].

The fluence of the prompt emission (i.e. the total time-integrated energy per unit area), lasting about 800s, was 0.013 $erg/cm^2$ in the 75–1000 keV energy range[23]. This value is a lower limit estimate, due to the partial saturation and pile-up caused by the intense GRB photon flux. As far as we know, this GRB is among the largest ever detected. Assuming both a measured distance corresponding to a red-shift z=0.151[27] and an isotropic emission (E-Iso) only in the high energy band, the energy emitted during the prompt GRB was about $8 \cdot 10^{53}$ ergs. It should be noted that both fluence and E-Iso values are lower limits, due to the partial saturation of instruments (SPI) and telemetry data transmission (IBIS). The prompt emission was followed by an unusual strong soft gamma-ray long tail[25] decaying with a power index around -2, and lasting at least 40 minutes before crossing the detection threshold of both SPI and IBIS detectors (Fig. 2).

GRB221009A strongly perturbed the D-region[13] (about 60–100 km of altitude) and, for the first time, its effect was observed also in the top-side ionosphere (507 km) by the Electric Field Detector (EFD)[28] onboard the Low Earth Orbit (LEO) Chinese Seismo Electromagnetic Satellite (CSES - see CSES satellite electric field data section for more details)[29], which was orbiting from North to South over the European sector (blue line in Fig. 1). Figure 3 shows the comparison between the SPI/ACS gamma-ray flux (panel a) and the ionospheric electric field measured by EFD (panel b). At 13:17:01 UT, EFD was switched on just before entering the Auroral Oval (AO). In this region (blue-shaded region), the electric field variations are strongly dominated by the auroral electrojet (a large horizontal current flowing mainly in the E

region of the ionosphere, located at an altitude of about 100 − 150 km), generated by complex solar wind-magnetosphere interaction processes[30–32]. This effect results in the impossibility to correlate the evolution of GRB221009A peaks from 13:17:07 to 13:20:44 UT. The position of the AO boundaries were determined by using Ding et al. algorithm[33]. At 13:25:03, about 1.5 min ($\Delta t_{GI}$) after the beginning of the third and final peak of GRB221009A, the EFD observed a strong peak in the ionospheric electric value of about $54 mV/m$. We hypothesize that such an electric field variation in the top-side ionosphere can be driven by the GRB221009A occurrence.

In fact, $\Delta t_{GI}$ might be related to a characteristic feature of the ionosphere in response to ionizing flux[34,35], which in general, depends on the balance between the electron production rate (dominated by photo-ionization) and the electron losses (resulting from recombination)[34,36,37]. The physical effect caused by the electron loss process is to delay the response of the changes in electron density $\rho_e$ to changes induced by the photo-ionization process. As a consequence $\Delta t_{GI}$ should represents the time taken for the ionospheric photo-ionization recombination processes to recover the equilibrium after an increase of irradiance. The higher is the ionospheric density, the larger is the delay time[35,37,38].

Figure 4 shows the CSES electric field observations during the GRB221009A occurrence for the three geographical components $E_x$ (panel a), $E_y$ (panel b) and $E_z$ (panel c), where x is directed northward, y westward, and z along the (negative) radial direction. It can be easily seen that the EFD variation (black curve) is superimposed to a low-frequency modulation induced by $v_s \times B$ effect[28].

At 13:25:03 UT a large peak in the ionospheric electric field is visible along the three components, whose amplitudes are: $\Delta E_x = 32.6 mV/m$; $\Delta E_y = -39.5 mV/m$; $\Delta E_z = 27.9 mV/m$.

## Discussion

These observations are consistent with an anomalous high ionization in the ionosphere. In general, such a ionospheric perturbations are caused by solar flares and/or solar particle events leading to sudden radio wave absorption (in both the medium frequency - MF - and high frequency - HF - ranges)[39]. These effects are detected in the D-region and are called Sudden Ionospheric Disturbances (SID)[40]. In the present case, the very strong and long-lasting photon flux due to GRB221009A

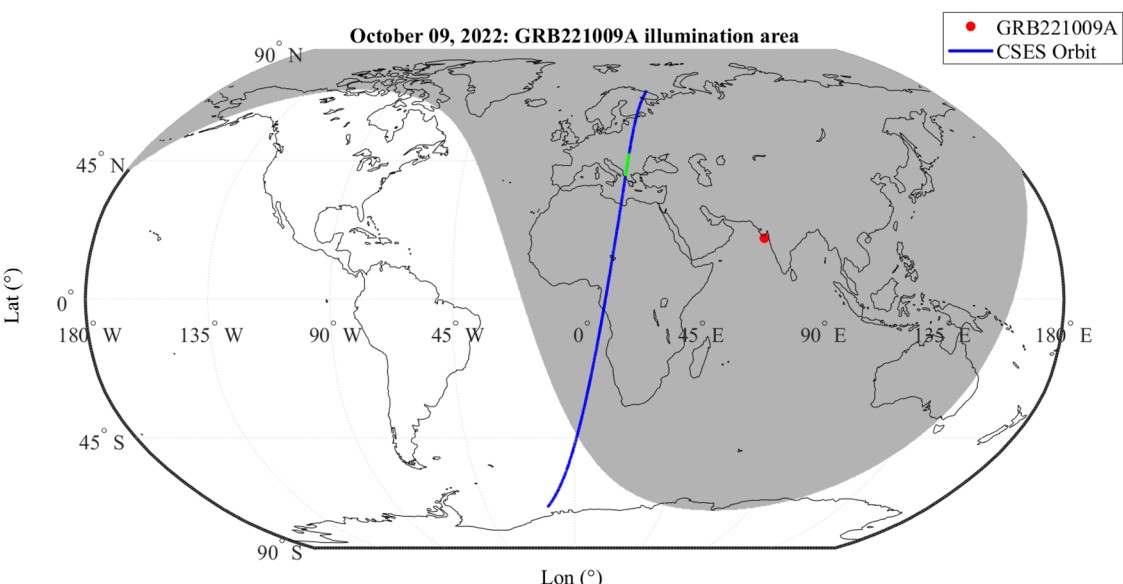

**Fig. 1 | Superposition of the GRB illumination area and CSES satellite orbit. A** map of the Earth with the CSES satellite orbit trace shown in blue. The green-colored part along the orbit marks the time of the electric field variation triggered

by the GRB and detected by EFD. The gray shaded area shows the estimated illumination area of GRB221009A impinged at a latitude of 19.8˙ and a longitude of 71˙ (red circle).

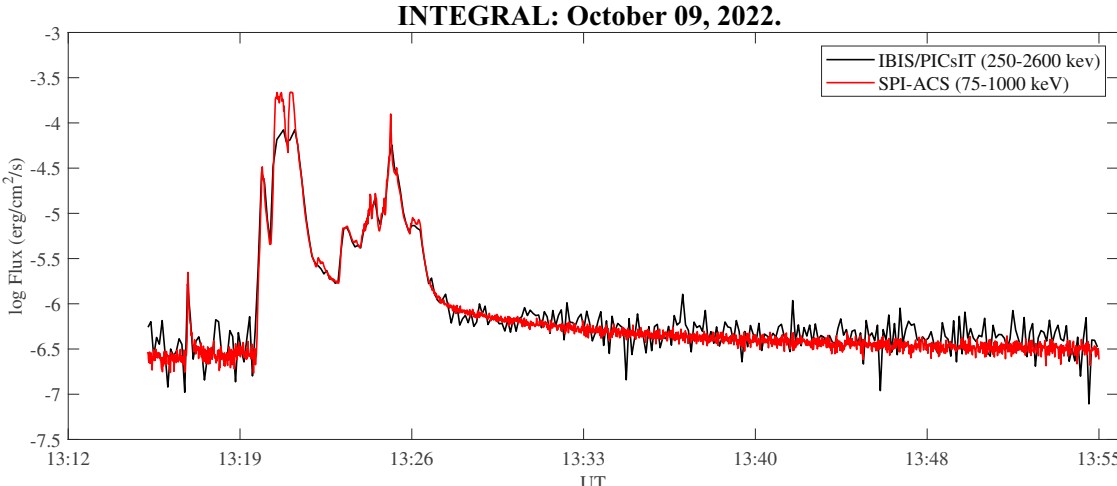

**Fig. 2 | Light curves from INTEGRAL satellite observations.** Time profile of *GRB*221009*A* detected by INTEGRAL, scaled for background. The red curve shows the SPI/ACS count rate on 1s time-bin plotted in *erg/cm²/s* in the energy range 75–1000 keV; the black curve shows the IBIS/PICsiT data in the energy range 0.25–2.60 MeV. The differences between the two light curves are due to: i) difference in computing the two energy bands, ii) statistical fluctuations (IBIS/PICSiT is less sensitive in this case because of the partial shield absorption to low energy photons), iii) instrument saturation and/or telemetry loss due to the exceptionally strong photon flux from GRB221009A.

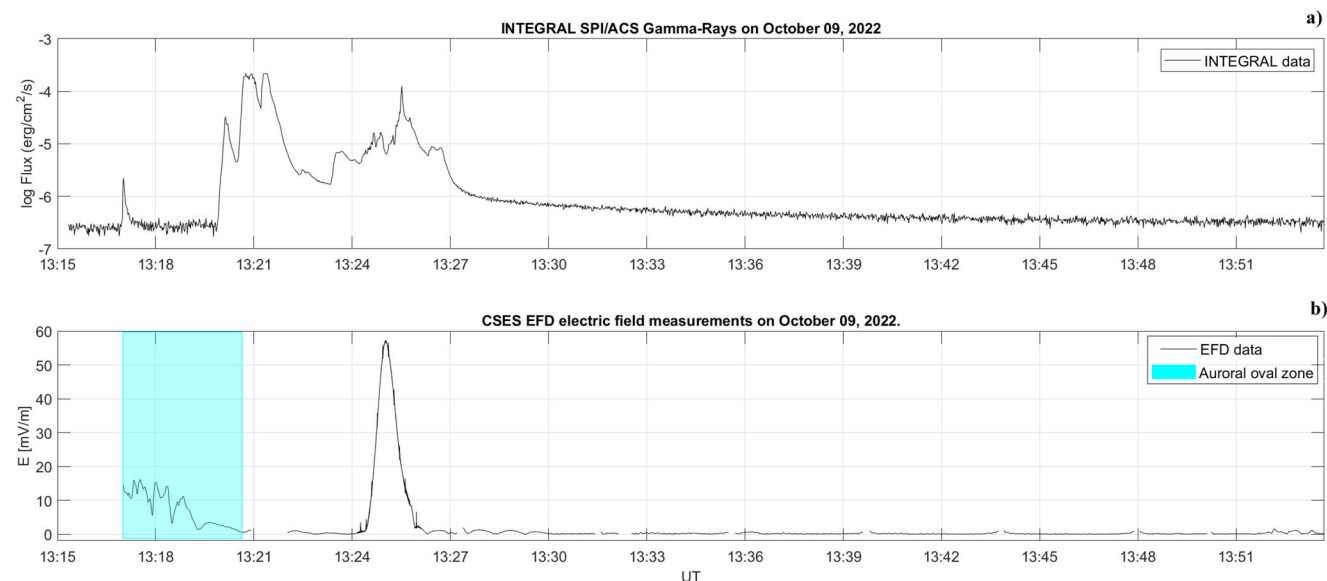

**Fig. 3 | Comparison of INTEGRAL gamma-ray and CSES electric field measurements.** Comparison between the SPI gamma-ray flux (panel **a**) and the ionospheric electric field observed by the CSES satellite (panel **b**), after the subtraction of the $v_s \times B$ induced electric field ($v_s$ and $B$ are the spacecraft speed and the local magnetic field, respectively). The blue-shaded region corresponds to the CSES flight over the Auroral Region. The CSES electric field observations were measured at an altitude of 507 km.

triggered an unprecedented level of ionization in the ionosphere producing both a significant SID in the bottom-side ionosphere and a strong electric field variation in the top-side ionosphere.

We hypothesize that the strong variations of the ionospheric electric field measured by CSES at an altitude of 507 km can only originate from a strong variation in the ionospheric parallel conductivity ($\sigma_0$)[41,42], which is directly dependent on the plasma density (see equation (5) in Analytical model for top-side Ionospheric Electric field variation induced by a GRB section). To confirm such a scenario, on the one hand we investigated the distribution of the ionospheric Total Electron Content (TEC) over Europe, as measured by Global Navigation Satellite System (GNSS) receivers (see GNSS Total Electron Content Data section). As from Fig. 5, GNSS receivers located in the Mediterranean area recorded a significant TEC increase on October 9th (panel b) between 13:00 and 14:00 UT compared to the day before (panel a) and after (panel c) at the same time, thus confirming the ionizing effect of the intense GRB[1,13,43].

On the other hand, we developed an analytical model (see Analytical model for top-side Ionospheric Electric field variation induced by a GRB for more details) able to give a first rough quantitative evaluation of the top-side ionospheric electric field variation driven by an impulsive photon flux (e.g., the impinging of a GRB). As can be seen from Fig. 6, an impulsive photon source can generate a variation in the top-side ionospheric electric field of about $30 mV/m$ only if the ratio $R_{\alpha\beta}$ between ion production ($\alpha$) and absorption ($\beta$) rates are greater than 5. In addition, if $R_{\alpha\beta}$ is lower than 2, the effect of the ionization seems not

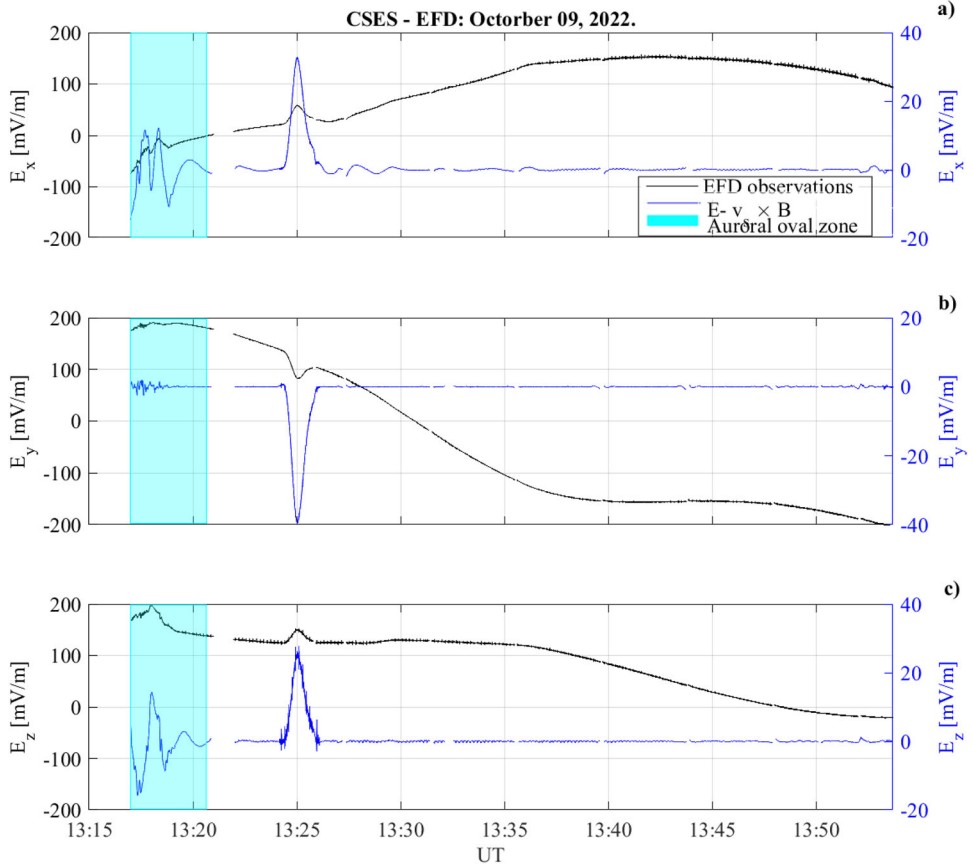

**Fig. 4 | Ionospheric Electric field observations from CSES satellite.** CSES electric field waveform as function of time during the GRB221009A occurrence for the three components $E_x$ (panel **a**), $E_y$ (panel **b**) and $E_z$ (panel **c**) with (black curve) and without (blue curve) the $v_s \times B$ induced electric field component. The blue-shaded region corresponds to the CSES flight over the Auroral Region. The CSES electric field observations were measured at an altitude of 507 km.

to be able to produce significant variation in the electric field. Such a result is in agreement with the previous experimental observations related to GRBs impinging the ionosphere[1,4,43,44].

In addition, our model predicts a time delay ($\Delta t_{th}$) between the peak of the GRB and the peak of the ionospheric electric field variation of 1.22 minutes for $R_{\alpha\beta} = 5$. This $\Delta t_{th}$ is in agreement with the $\Delta t_{Gl}$ observed.

As previously said, being the observations in the bottom-side ionosphere analogous to the effects induced by solar flares (Solar Flare Effect - SFE)[45,46], we investigated the possibility of a sudden intensification of the Solar quiet (Sq) ionospheric current system[47,48] and of the ionospheric Equatorial Electrojet (EEJ)[49,50] induced by the GRB221009A[51]. The Sq ionospheric electric currents are located in the E-region and are responsible of the diurnal variation in the geomagnetic field observed at ground[52]. Figure 7b shows the comparison between the equatorial electrojet, estimated in terms of the variation of the North-South component of the geomagnetic field (H - see Equatorial electrojet evaluation section for more details), calculated for a solar quiet day (October 12$^{th}$, 2022, black line) and for the day of the GRB occurrence (October 9$^{th}$, 2022, red line). It can be seen that the occurrence of the GRB221009A (vertical black dashed line) generated a perturbation of the EEJ. Indeed, superimposed to the long-term variation, featured in both days and characterized by a minimum around both dawn and dusk, and by a maximum around the noon, at about 13:21 UT a low frequency (0.35 mHz) fluctuation appears. Such a variation is more clear in the original magnetometer data used for the EEJ evaluation (panels a, b, c and d) in Fig. 7). In fact, looking at Tatuoka data (panels b and d) which is located inside the EEJ, we can see that during quiet conditions (panel b) the geomagnetic field reaches its

maximum values around the local noon remaining almost stable for about 2.5 hours before decreasing down as the station approaches the local dusk[49]. Differently, on October 9th (panel d), before the GRB occurrence, as expected the H field reaches its maximum value, but, around 13:21 UT, in coincidence with the occurrence of the first peak of the GRB (vertical black dashed line), instead of remaining stable, starts to fluctuate with a low frequency of 0.35 mHz. Interestingly, such alteration lasted up to 19:00 UT, possibly sustained by the hard GRB221009A long tail (Fig. 2), containing more than 10% of the total energy of the prompt emission.

In conclusion, the unprecedented photon-flux associated to the GRB221009A deeply impacted on the Earth's ionospheric conductivity, causing a strong perturbation not only in the bottom side ionosphere[13,14], where it is typically observed using ground VLF antennas[53], but also in the top-side ionosphere (at around 500 km). In fact, a huge variation of the ionospheric electric field, induced by the strong ionospheric conductivity change was detected in the top side ionosphere (507 km) as a consequence of a GRB impact, which increased the ionospheric plasma density by the huge photo-ionization (even in the dayside), as depicted in Fig. 5. The analytical model described in this work supports the observations and confirms the hypothesis that the interaction between GRB and top-side ionosphere is a threshold process[1,4,44]. Our model suggests that such a threshold strictly depends on both the production-to-loss-rate ratio of ions and the time duration of the ionization process.

As a closing remark, we want to highlight that, differently to previous similar studies[13] focused on the impact of GRB on both D- and F- regions by using TEC data[6,43] and/or VLF ground electromagnetic transmitters[1,4,14], our work represents, at our knowledge, the first-ever

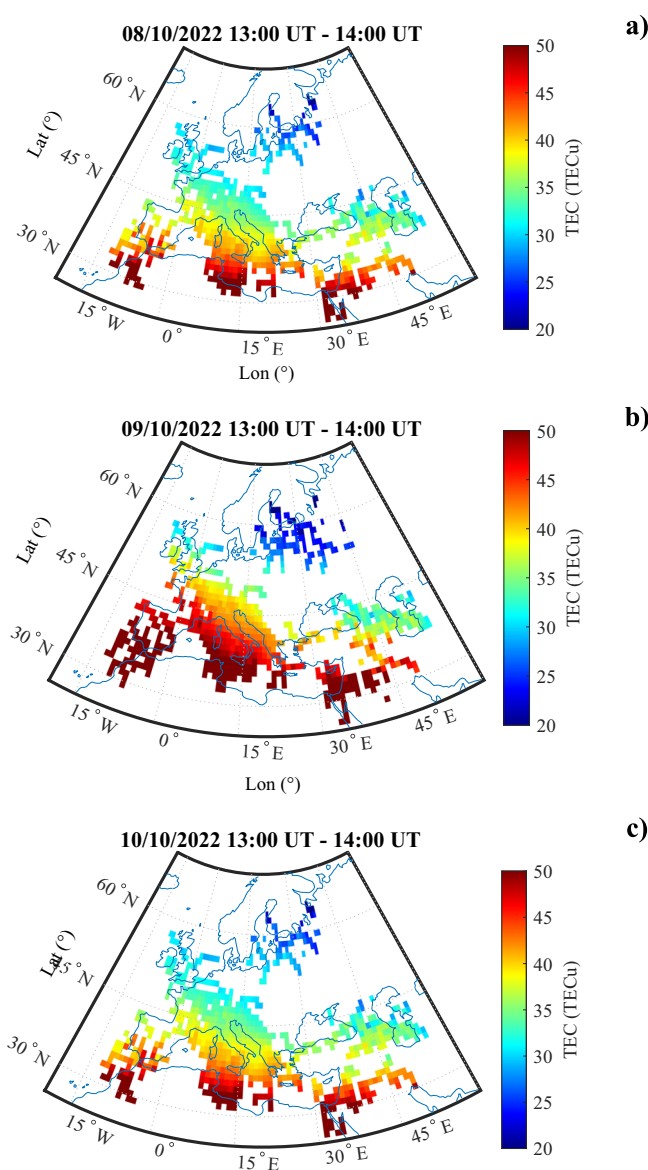

**Fig. 5 | TEC map over Europe during the GRB occurrence.** Map of the vertical total electron content (TEC) around the CSES satellite position one day before (panel **a**), at the moment of (panel **b**) and one day after (panel **c**) the GRB occurrence. All the maps have been averaged over 1 hour, between 13:00 UT and 14:00 UT. Colors are representative of the TEC value.

top-side ionospheric (507km) measurement of electric field variation triggered by impulsive cosmic photons.

## Methods

This section contains the description of the datasets used in this study and the analytical description of the model developed for the explanation of the experimental results.

### INTEGRAL satellite data

INTEGRAL, an ESA lead space observatory for observations in the energy range from a few keV up to 10 MeV, was launched in 2002 and is still fully operative. In this study data from the Imager IBIS[54] and the SPectrometer SPI[55] have been used. In particular, IBIS observes from 25 keV to 10 MeV, with an angular resolution of 12 arcmin, enabling a bright source to be located to better than 1 arcmin. SPI observes

radiation between 20 keV and 8 MeV with an high energy resolution of 2 keV at 1 MeV, capable to resolve candidate gamma-ray lines[56]. The INTEGRAL instruments were pointed to a sky direction at about 60 degree offset respect to the GRB arrival direction and the signal were detected by the omni-directional SPI/ACS shield and by the IBIS/PICsIT detector through the telescope shield (see annex material for detailed telescope response[57]). The INTEGRAL data are transmitted continuously in real-time to ground, and distributed in almost real-time via GCN web network and also through the Interplanetary Network (IPN).

### CSES satellite electric field data

CSES-01 (Chinese Seismo-Electromagnetic Satellite) is a LEO satellite orbiting sun-synchronously at about 507 km since February 2018[29,58]. CSES-01 has nine instruments on board for the electromagnetic field, waves and charged particle observations in the upper ionosphere. For this analysis we used electric field data from the Electric Field Detector (EFD)[59]. EFD is able to measure the electric field in four frequency bands: ULF (DC -16 Hz) with a sampling frequency of 125 Hz; ELF (6 Hz–2.2 kHz) with a sampling frequency of 5 kHz; VLF (1.8 kHz–20 kHz) with a sampling frequency of 50 kHz; and HF (High-frequency, 18 kHz–3.5 MHz) with a sampling frequency of 50 kHz. Due to the limitation of telemetry capability, the waveform data are only available for both the ULF and ELF bands, and for a few minutes, over the global seismic belts, for both VLF and HF bands. During the remaining part of the orbit the VLF and HF data are transmitted as Fast Fourier Transform (FFT)[28].

To eliminate the $v_s \times B$ effect ($v_s$ and $B$ are the spacecraft speed and the local magnetic field, respectively), induced by the motion of the satellite inside the geomagnetic field, from the E field components, we applied the technique described in Diego et al.[28].

### GNSS total electron content data

To investigate the ionospheric scenario leading to the observed impulsive variation of the current generated in the ionosphere, we collected and processed standard daily RINEX files provided by the permanent stations, located in Europe, of the University NAVSTAR Consortium and of the Rete Integrata Nazionale GNSS[60] managed by the Istituto Nazionale di Geofisica e Vulcanologia (INGV). In particular, to calibrate vertical total electron content (vTEC) data, we processed GNSS measurements as described in D'Angelo et al.[61] by using the technique by Ciraolo et al.[62] and Cesaroni et al.[63]. Specifically, to generate maps over a specific world zone, we performed an average of one hour vTEC observations over $1° \times 1°$ of geographic latitude and longitude bin using data recorded by all satellites in view of each selected GNSS ground receiver.

### Equatorial electrojet evaluation

The equatorial electrojet (EEJ) was obtained using the method described in Soares et al.[64]. We considered the H (North-South) component the geomagnetic field at ground alone, being directly related to the east-west flow of the EEJ[49]. We used two pairs of ground stations consisting of one magnetometer close to the magnetic equator and one out at almost the same meridian. This assumption allows to have only one observatory under the influence of the EEJ. To estimate the EEJ, we evaluated the difference between the H component measured by the two pair stations after the subtraction of the nighttime baseline. Finally the EEJ signal at the longitude of the equatorial stations is obtained referred to as $\Delta H$. The ground stations information used for the EEJ estimation are reported in Table 1.

Magnetometer data were obtained from INTERMAGNET magnetometer array network. INTERMAGNET is a consortium of observatories and operating institutes that guarantees a common standard of data released to the scientific community, allowing the possibility to compare the measurements carried out at different observation points.

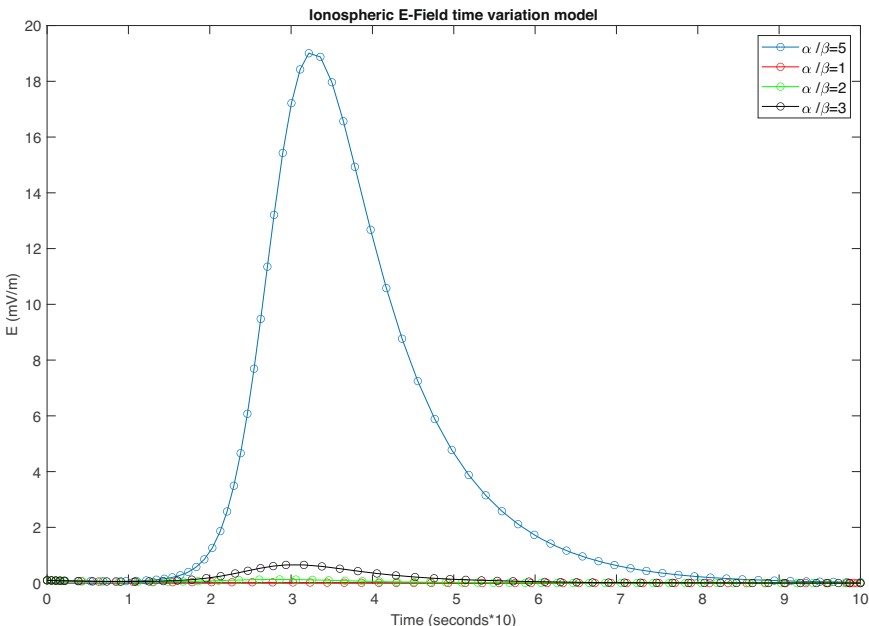

**Fig. 6 | Modelling of ionospheric electric field variation induced by a GRB.** Model results of top-side ionospheric electric field time variation induced by a impulsive photon source. Colours are representative of different photon production/absorption rate ratio.

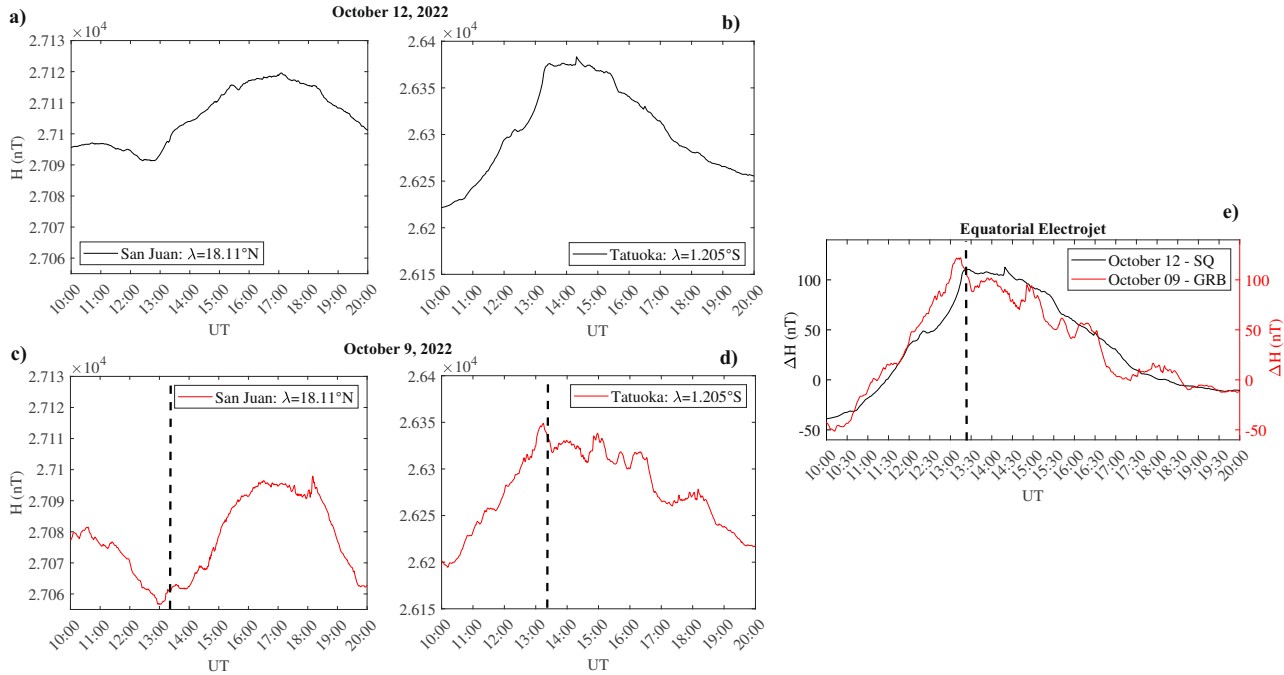

**Fig. 7 | Comparison between Equatorial Electrojet during quiet period and GRB occurrence.** Estimation of the Equatorial Electrojet for the a solar quiet day of October 2022 (black line) and for the day of the GRB occurrence (red line): panel **a** and **c** show original observations of the H component of the geomagnetic field from San Juan magnetometer station during quiet and GRB day, respectively; panel **b**) and **d**) show original observations of the H component of the geomagnetic field from Tatuoka magnetometer station during quiet and GRB day, respectively; panel **e**) shows the EEJ results in terms of $\Delta H$. Black dashed lines represent the time occurrence of the first peak of the GRB.

## Analytical model for top-side Ionospheric Electric field variation induced by a GRB

In order to develop a model able to represent the effect of GRB impinging the top-side ionosphere, we started from the ionospheric Ohm's law[65]:

$$\mathbf{J} = \boldsymbol{\sigma} \cdot \mathbf{E} = \sigma_0 \mathbf{E}_{||} + \sigma_p \mathbf{E} + \sigma_H \frac{\mathbf{B} \times \mathbf{E}}{\mathbf{B}}, \qquad (1)$$

where E is the electric field, B is the ambient magnetic field, σ is the conductivity tensor with $\sigma_p$, $\sigma_H$, and $\sigma_0$ being respectively the Pedersen, Hall, and parallel conductivity. The formation of the electric current in the ionized layer is caused by the difference between the velocities of ions (typically $NO^+$, $O_2^+$, $O^+$, $H^+$, $H_e^+$ and $N^+$) and electrons. In ionosphere, the temporal variability of the electrodynamics processes is slow enough that one can ignore the displacement current in Maxwell's equations (i.e., the term $\partial E/\partial t$)[41], therefore Ampère-Maxwell law

**Table 1 | Magnetometer Ground Station Location: Information about the location of the ground magnetometer stations used for the EEJ estimation**

| Station | IAGA Code | Latitude (°) | Longitude (°) |
|---------|-----------|--------------|---------------|
| San Juan | SJG | 18.11 N | 293.85 E |
| Tatuoca | TTB | -1.205 N | 311.487 E |

reduces to

$$\mu_0 \mathbf{\nabla} \cdot \mathbf{J} = \mathbf{\nabla} \cdot (\mathbf{\nabla} \times \mathbf{B}) = 0, \tag{2}$$

where $\mu_0$ is vacuum magnetic permeability. By combining equations (1) and (2), we obtain:

$$\mathbf{\nabla} \cdot (\boldsymbol{\sigma} \cdot \mathbf{E}) = 0. \tag{3}$$

At about 500 km (i.e. CSES orbiting altitude) both $\sigma_H$ and $\sigma_p$ are negligible with respect to $\sigma_0$ (see Fig. 7 in Denisenko et al.[66]). As a consequence, equation (3) simplifies to

$$\mathbf{\nabla} \cdot (\boldsymbol{\sigma_0} \cdot \mathbf{E}_{||}) = 0. \tag{4}$$

Once $\sigma_0$ is known, equation (4) can be numerically solved to obtain the $E$ field behaviour. Equation for the parallel conductivity in the ionosphere as given by Maeda[42] reads

$$\sigma_0 = \frac{n_e q_e^2}{m_e \nu_e} \tag{5}$$

$$\nu_e = \nu_{e,i} + \nu_{e,n}, \tag{6}$$

where $n_e$ is the electron density, $\nu_{e,n}$ is electron-neutral collision frequency, $\nu_{e,i}$ electron-ion collision frequency, $q_e$ is the unsigned electric charge (i.e. $1.602 \cdot 10^{-19} C$), and $m_e$ is the electron mass (i.e. $9.109 \cdot 10^{-31} kg$). Following the results of Aggarwal et al.[67], we can estimate the electron collision frequency at about 500 km as $\nu_e = 10^2 \text{sec}^{-1}$.

Being $\sigma_0$ directly dependent on the electron density, it is straightforward that any variation of $n_e$ causes a changes in E. In general, the rate of change of the electron density is expressed by a continuity equation[68]:

$$\frac{dn_e}{dt} = A - L, \tag{7}$$

where $A$ is the production coefficient and $L$ the loss coefficient by recombination/losses. Naturally, the recombination coefficient depends of what ion species are present, and hence on the ionospheric altitude. At high altitudes (>200km, i.e. top-side ionosphere) where $O^+$ is the dominant ion species, $L$ becomes proportional to the electron density[68]. So, equation (7) becomes:

$$\frac{dn_e}{dt} = A - \beta n_e, \tag{8}$$

where $\beta$ is the loss rate. Equation (8) is valid only at altitudes higher than 200 km (and hence at the altitude of our electric field observations), being $L$, at lower altitudes (bottom-side ionosphere), proportional to the square of the electron density[68]. To simulate the production rate induced by a GRB, we used a Gaussian impulsive

function of the form $\alpha e^{-\left(\frac{t-t_0}{s_0}\right)^2}$, so that equation (8) can be written as:

$$\frac{dn_e}{dt} = \alpha e^{-\left(\frac{t-t_0}{s_0}\right)^2} - \beta n_e, \tag{9}$$

where $\alpha$ is the production rate induced by the GRB that depends on its photon flux, $t_0$ is the time of the maximum production rate, and $s_0$ is the width of the pulse.

Putting together equations (4), (5), and (9), and assuming at 500 km both an average electron density of $1.2 \cdot 10^{11} \text{cm}^{-3}$ [69] and an average loss rate coefficient of $0.6 \cdot 10^{-6} \text{sec}^{-1}$ [70–72], we can model the electric field variation induced by a GRB as a function of the ionospheric plasma density variation at 500 km of altitude. Figure 6 shows the results of our model for different ratios between production ($\alpha$) and loss ($\beta$) rate. It can be easily seen that the effect of a GRB is negligible if $\alpha/\beta < 3$. To obtain results similar to what was observed on October 9th, 2022, our model requires a production-to-loss ratio greater than 5.

The usage of a formalism directly related to the ratio between $\alpha$ and $\beta$ allows the model to being independent (for the present analysis) of the calculation of a realistic photon production rate caused by a GRB, whose evaluation needs a Montecarlo approach and the estimation of the real top-side ionospheric ion cross-section, which is out of the scope of the present work but a more accurate modelling of the effect of a GRB on the top-side ionospheric electric field is in progress.

Anyway, despite being very simplified, our model can be used to give a first quantitative explanation of the effect induced in the top-side ionosphere by GRB221009.

## Data availability

We cannot supply our source data in any public depository since they are property of: European Space Agency (INTEGRAL satellite data); Italian Space Agency (CSES satellite data); International Real-time Magnetic Observatory Network (ground magnetometer data); University NAVSTAR Consortium (GNSS satellite data). Anyway all of them can be freely downloaded from the relative website after registration. CSES satellite data are freely available at the LEOS repository (www.leos.ac.cn/#/home, accessed on 08/09/2023) after registration; GNSS data are freely available at University NAVSTAR Consortium (https://www.unavco.org/accessedon08/09/2023) after registrtion. INTEGRAL SPI data are freely available at the ISDC (https://www.isdc.unige.ch/integral/, accessed on 08/09/2023) repository. INTEGRAL PiCsIt/IBIS data are proprietary data of authors of the paper without any restriction. Ground magnetometer data are freely available at INTERMAGNET website (https://imag-data.bgs.ac.uk/GIN_V1/GINForms2, accessed on 08/09/2023). The datasets generated during and/or analysed during the current study are available from the corresponding author on request.

## Code availability

Codes used to produce results and figures were obtained using Matlab software package. They are not public but can be made available upon request to the corresponding author.

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

## Acknowledgements

The authors thank the Italian Space Agency for the financial support under the contract ASI "LIMADOU Scienza+" n° 2020-31-HH.0, and the financial support under the " INTEGRAL ASI-INAF" agreement n° 2019-35-HH.0. M.P., and G.d.A. thank the ISSI-BJ project "The electromagnetic data validation and scientific application research based on CSES satellite" and Dragon 5 cooperation 2020-2024 (ID. 59236). Part of the research leading to the result has receiving founding support from the EuropeanUnion's Horizon 2020 Programme under the AHEAD2020 project (grant agreement n. 871158). This material is based on services provided by the GAGE Facility, operated by EarthScope Consortium, with support from the National Science Foundation, the National Aeronautics and Space Administration, and the U.S. Geological Survey under NSF Cooperative Agreement EAR-1724794. The authors thank the INGV group for providing the RING data. We thank the national institutes that support INTERMAGNET for promoting high standards of the magnetic observatory practice (www.intermagnet.org) used in this paper.

## Author contributions

M.P. writing–original draft, formal analysis, methodology and supervision; P.U. writing–revision, editing and methodology; R.B. writing–revision, formal analysis and validation; A.B. writing–revision; G.d.A. writing–revision, formal analysis; J.C.R. writing–revision, formal analysis; P.D. data validation, funding, Z.Z data validation. R.A., D.B., S.B., S.Be., I.B., W.J.B., D.C., A.C., P.C., S.C., L.C., A.Co., M.C., F.d.A., C.d.D., C.d.S., A.d.L., E.F., F.M.F., G.G., R.I., A.L., M.L., B.M., G.M., M.M., M.Me., M.Mes, A.M., C.N, F.N., F.Nu, A.O., G.O., F.P., F.Pa., B.P., E.P., A.P., S.P., F.P., A.Pe., P.P., M.Po., G.R., D.R., E.R., M.R., S.B.R., A.R., X.S., Z.S., U.S., V.S., A.S., R.S., S.T., N.V., V.V., V.Vi, U.Z., S.Z., P.Z., are part of the CSES-Limadou Collaboration whose significant contribution made satellite observations possible.

## Competing interests

This research received no external funding. The authors declare no competing interests.

## Additional information

Mirko Piersanti [1,2,21] ✉, Pietro Ubertini [2,21], Roberto Battiston [3,4,21], Angela Bazzano[2,21], Giulia D'Angelo [2,21], James G. Rodi[2,21], Piero Diego[2,21], Zhima Zeren[5], Roberto Ammendola [6], Davide Badoni [6], Simona Bartocci[6], Stefania Beolè[7], Igor Bertello[2], William J. Burger[4], Donatella Campana[8], Antonio Cicone[2,9], Piero Cipollone[6], Silvia Coli[7], Livio Conti [6,10], Andrea Contin[11,12], Marco Cristoforetti [4,13], Fabrizio De Angelis[2], Cinzia De Donato[6], Cristian De Santis [6], Andrea Di Luca [3,4], Emiliano Fiorenza[2], Francesco Maria Follega [3,4], Giuseppe Gebbia [3,4], Roberto Iuppa[3,4], Alessandro Lega[3,4], Mauro Lolli[12], Bruno Martino[2,14], Matteo Martucci[6], Giuseppe Masciantonio [6], Matteo Mergè[6,15], Marco Mese[8,16], Alfredo Morbidini[2], Coralie Neubüser [4], Francesco Nozzoli [4], Fabrizio Nuccilli[2], Alberto Oliva [11,12], Giuseppe Osteria [8], Francesco Palma[6], Federico Palmonari[11,12], Beatrice Panico[8,16], Emanuele Papini[2], Alexandra Parmentier[2,6], Stefania Perciballi [7], Francesco Perfetto[8], Alessio Perinelli [3,4], Piergiorgio Picozza[6,17], Michele Pozzato[12], Gianmaria Rebustini[6], Dario Recchiuti [2,3], Ester Ricci [3,4], Marco Ricci[18], Sergio B. Ricciarini[19], Andrea Russi [2], Zuleika Sahnoun[12], Umberto Savino[7], Valentina Scotti [8,16], Xuhui Shen[20], Alessandro Sotgiu [6], Roberta Sparvoli[6,17], Silvia Tofani[2], Nello Vertolli[2], Veronica Vilona [3], Vincenzo Vitale [6], Ugo Zannoni[2], Simona Zoffoli [15] & Paolo Zuccon [3,4]

[1]Department of Physical and Chemical Sciences, University of L'Aquila, 67100 L'Aquila, Italy. [2]National Institute of Astrophysics, IAPS, Rome 00133, Italy. [3]Department of Physics, University of Trento, Povo, Italy. [4]TIFPA-INFN, Povo, 38123 Trento, Italy. [5]National Institute of Natural Hazards, Ministry of Emergency Management of China, Beijing 100085, People's Republic of China. [6]INFN, University of Rome Tor Vergata, Rome 00133, Italy. [7]INFN - Sezione di Torino, 10125 Torino, Italy. [8]INFN-Sezione di Napoli, Naples 80126, Italy. [9]Dipartimento di Ingegneria e Scienze dell'Informazione e Matematica, University of L'Aquila, 67100 L'Aquila, Italy. [10]Uninettuno University, 00186 Rome, Italy. [11]University of Bologna, Bologna 40127, Italy. [12]INFN - Sezione di Bologna, 40127 Bologna, Italy. [13]Fondazione Bruno Kessler, 38123 Povo, TN, Italy. [14]CNR, V. Fosso del Cavaliere 100, 00133 Rome, Italy. [15]Agenzia Spaziale Italia, Rome 00133, Italy. [16]Università degli Studi di Napoli Federico II, 80126 Naples, Italy. [17]Department of Physics, University of Rome Tor Vergata, Rome 00133, Italy. [18]INFN-LNF, Frascati, Rome 00100, Italy. [19]IFAC-CNR, Sesto Fiorentino, Florence 50019, Italy. [20]National Space Science Center, Chinese Academy of Sciences, Beijing 100190, People's Republic of China. [21]These authors contributed equally: Mirko Piersanti, Pietro Ubertini, Roberto Battiston, Angela Bazzano, Giulia D'Angelo, James G. Rodi, Piero Diego. ✉e-mail: mirko.piersanti@univaq.it

