## [Peer Review File · Nature Communications]

Evidence of an upper ionospheric electric field perturbation correlated with a gamma ray burstREVIEWER COMMENTS

Reviewer #1 (Remarks to the Author):

Summary:

Gamma-ray bursts are known to have an impact on the lower ionosphere of the Earth. In this manuscript the authors investigated the possible impacts of a very large gamma-ray burst (GRB) on the Earth's upper ionosphere.

GRB impacts on the upper ionosphere have not been observed to date, and it is not clear how GRBs could directly or indirectly produce disturbances in this region.

The authors of this work report variations in ionosphere electric fields observed by the CSES satellite which was in a low Earth orbit over the side of the Earth that was illuminated by a particularly large GRB, which occurred on October 9. This is an interesting and significant result, but their physical explanation for how the D-region and F-region are linked is not well described and is not convincing.

In addition, the authors describe total electron content (TEC) and estimate the equatorial electrojet, but it is not clear from these data (Figure 3 and 4) if they effects are significant and therefore if they bare any connection to the SID created by the GRB in the D-region.

Minor concern: The paper does not flow well and is challenging to read due to its poor structuring. I would encourage the authors to better organise the paper so that it is easier of the reader to appreciate their observations and results.

While the results are of potential interest, the data are not well presented and the physical link between the D- and F-regions is not convincing.

I therefore recommend that this manuscript is revised significantly.

Additional comments below:

- "Earth's atmosphere ... plays a fundamental role for the evolution and endurance of human life".

Not limited to human life. All life depends on Earth's atmosphere and its stability.

- "... effects on the ionosphere were rarely observed, in any case on its bottom-side."

Not clear what this means. Please rephrase.

- "... first evidence of an intense top-side ionospheric perturbation ... induced by a very significant Sudden Ionospheric Disturbance".

Maybe a problem with sentence structure, but could be interpreted as authors saying that the top-side perturbation is caused by a SID.

Main:

- Acronym should be defined for spacecraft and instruments when first mentioned in text. e.g., OSIRIS, GTC, etc.

- Figure 6 is poor quality. Not clear where double ionisation event is. Maybe use Mercator projection. Hard to discern continents. No labels for latitude and longitude. Should label CSES orbits in figure.

- Figure 1: Use one common time axis for both panels. "CSES vs INTEGRAL" in title is not what is shown in the figure. Title should be something like "INTEGRAL SPI/ACS Gamma-Rays" on top title and "CSES Electric Fields" below. Should label blue shaded region in figure. "Time (UT)" should be along bottom axis. What altitude were the CSES measurements made at?

- Define heights of "bottom-side" and "top-side" when first used.

- Add references and possibly web addresses for the following: "Electric Field Detector (EFD) aboard the Low Earth Orbit Chinese Seismo Electromagnetic Satellite (CSES)".

- Figure 2: Use common time axis. Label black and blue curves in figure. Also label shaded region. What altitude was CSES at when the E-field measurements were made?

- "GNSS receivers located in the Mediterranean area recorded a significant TEC increase on October 9 between 13:00 and 14:00 UT compared to the day before and after at the same time, confirming the ionizing effect of the intense GRB28,40."

It is not clear from this figure that there is a significant increase in TEC on October 9. Suggest zooming in on the region of interest and/or labelling the region of interest. Also, the authors should quantify what the percentage increase was in TEC. Note that the colour bar unit is "TECu" while the caption states "vTEC". Please define.

The authors state: "we believe that the strong variations of the ionospheric vertical component (E_z) of the electric field measured by CSES satellite can only originate from a strong sudden ionization of the layer above the D-region" and then go not to discuss TEC measurements using GNSS receivers. It is not clear how these are related in this paragraph.

- "E-layer": Define and give height range. Same for D- and F-regions when first mentioned.

- "SQ": Define.

- Figure 4 show the equatorial electrojet estimated from the North-South component of the geomagnetic field (ΔH). Is H is horizontal component of the magnetic field? Needs to be defined. The authors should show the original H-field data, together with the variation in the N-S component of the field (ΔH). Also, it would be useful to remind readers of when the gamma-ray burst occurred, by over plotting it on this figure.

- The last paragraph of the paper is completely unrelated to the main focus of the work:

"Potentially, strong GRB events might abruptly deplete stratospheric ozone on a global scale³. Thomas et al. (2005)⁴ estimated that a global Ozone variation of 35% caused by a strong GRB or a similar cosmic event (statistically occurring at least once in a billion of years) would trigger widespread extinction. In fact, during the recovery phase of the Ozone (← 5 years) the Earth's surface would be exposed to up to a 3-fold increase in solar UVB flux, representing the most dangerous hazard for the biological system. Therefore, it is crucial to investigate how a strong-long lasting GRB impacts and modifies the Earth's atmosphere whose ionization stability plays a fundamental role for the evolution and endurance of human life¹."

Reviewer #2 (Remarks to the Author):

This is a straightforward report of an extragalactic event modifying the Earth's atmosphere. It is important and well-written. I unreservedly recommend its publication.

Reviewer #3 (Remarks to the Author):

Paper #: NCOMMS-23-05263-T

Title: First Evidence of Earth's top-side ionospheric electric field variation triggered by impulsive cosmic photons

Authors: Piersanti et al.

Summary: This paper presents evidence of an electric field enhancement observed onboard the CSES satellite (~500 km altitude) shortly after the occurrence of the gamma ray burst on October 9, 2022. The authors suggest these observations are the first evidence of an F-region effect by gamma-ray bursts. Furthermore, they suggest that these observations are evidence of a large sudden ionospheric disturbance at lower altitudes. They present GPS TEC observations that show larger

electron densities over some regions of Europe on the day of the gamma ray burst. The electric field observations are convincing, but the working theory explaining the properties of the electric field is not clear, and they do not appear to be the first reported F-region effect of gamma ray bursts. It is also not clear why the electric field enhancement is necessarily associated with a sudden ionospheric disturbance at lower altitudes. Lastly, it is not clear why the TEC enhancements over Europe are larger in some areas and smaller in others, especially when the gamma ray burst is centered over India. Overall, this paper requires significant clarification prior to publication.

Detailed Comments:

The evidence of a large electric field enhancement occurring about 40 seconds after the third large gamma ray burst peak is solid. The description of the physical process that might explain why the electric field enhancement occurs with such a significant delay requires further explanation. The explanation of the relatively short duration of the electric field enhancement compared to the duration of the gamma ray burst also requires further explanation. Without these explanations, I do not understand the relationship between the electric field enhancement and a sudden ionospheric disturbance at lower altitudes.

The TEC observations indicate an increase in TEC over Italy during the event, but not over any other locations. For instance, over the Caucasus it seems that the TEC is unaffected by the gamma ray burst. This is unexpected for a gamma ray burst centered over India. Is this the case, or is the TEC measurement less sensitive in these locations?

How much variation is observed on the magnetometer observations on a daily basis? For instance, the deviation from the quiet day curve appears to occur well before the gamma ray burst precursor. Why should the variations that occur before the event be neglected, but the variations that occur after the event be ascribed as an effect of the gamma ray burst?

Lastly, some modification is required to the statement that this is the first evidence of an F-region effect of a gamma ray burst. Mahrous (2017, citation below) shows TEC changes associated with a gamma ray burst. TEC is dominated by F-region electron density, so it would appear that Mahrous has demonstrated an F-region effect of a gamma ray burst. The authors should address this article. (I have not seen evidence of electric field enhancements associated with gamma ray bursts in previous publications, however.)

Mahrous, A. (2017), Ionospheric response to magnetar flare: signature of SGR J1550--5418 on coherent ionospheric Doppler radar, *Annales Geophysicae*, 35 (3), p.345-351, DOI: 10.5194/angeo-35-345-2017.

REVIEWER COMMENTS

Reviewer #1 (Remarks to the Author):

Summary:

Gamma-ray bursts are known to have an impact on the lower ionosphere of the Earth. In this manuscript the authors investigated the possible impacts of a very large gamma-ray burst (GRB) on the Earth's upper ionosphere.

GRB impacts on the upper ionosphere have not been observed to date, and it is not clear how GRBs could directly or indirectly produce disturbances in this region.

The authors of this work report variations in ionosphere electric fields observed by the CSES satellite which was in a low Earth orbit over the side of the Earth that was illuminated by a particularly large GRB, which occurred on October 9. This is an interesting and significant result, but their physical explanation for how the D-region and F-region are linked is not well described and is not convincing.

In addition, the authors describe total electron content (TEC) and estimate the equatorial electrojet, but it is not clear from these data (Figure 3 and 4) if they effects are significant and therefore if they bare any connection to the SID created by the GRB in the D-region.

Minor concern: The paper does not flow well and is challenging to read due to its poor structuring. I would encourage the authors to better organise the paper so that it is easier of the reader to appreciate their observations and results.

While the results are of potential interest, the data are not well presented and the physical link between the D- and F-regions is not convincing.

I therefore recommend that this manuscript is revised significantly.

Additional comments below:

- "Earth's atmosphere ... plays a fundamental role for the evolution and endurance of human life".

Not limited to human life. All life depends on Earth's atmosphere and its stability.

- "... effects on the ionosphere were rarely observed, in any case on its bottom-side."

Not clear what the this means. Please rephrase.

- "... first evidence of an intense top-side ionospheric perturbation ... induced by a very significant Sudden Ionospheric Disturbance".

Maybe a problem with sentence structure, but could be interpreted as authors saying that the top-side perturbation is caused by a SID.

Main:

- Acronym should be defined for spacecraft and instruments when first mentioned in text. e.g., OSIRIS, GTC, etc.

- Figure 6 is poor quality. Not clear where double ionisation event is. Maybe use Mercator projection. Hard to discern continents. No labels for latitude and longitude. Should label CSES orbits in figure.

- Figure 1: Use one common time axis for both panels. "CSES vs INTEGRAL" in title is not what is shown in the figure. Title should be something like "INTEGRAL SPI/ACS Gamma-Rays" on top title and "CSES Electric Fields" below. Should label blue shaded region in figure. "Time (UT)" should be along bottom axis. What altitude were the CSES measurements made at?

- Define heights of "bottom-side" and "top-side" when first used.

- Add references and possibly web addresses for the following: "Electric Field Detector (EFD) aboard the Low Earth Orbit Chinese Seismo Electromagnetic Satellite (CSES)".

- Figure 2: Use common time axis. Label black and blue curves in figure. Also label shaded region. What altitude was CSES at when the E-field measurements were made?

- "GNSS receivers located in the Mediterranean area recorded a significant TEC increase on October 9 between 13:00 and 14:00 UT compared to the day before and after at the same time, confirming the ionizing effect of the intense GRB28,40."

It is not clear from this figure that there is a significant increase in TEC on October 9. Suggest zooming in on the region of interest and/or labelling the region of interest. Also, the authors should quantify what the percentage increase was in TEC. Note that the colour bar unit is "TECu" while the caption states "vTEC". Please define.

The authors state: "we believe that the strong variations of the ionospheric vertical component (E_z) of the electric field measured by CSES satellite can only originate from a strong sudden ionization of the layer above the D-region" and then go not to discuss TEC measurements using GNSS receivers. It is not clear how these are related in this paragraph.

- "E-layer": Define and give height range. Same for D- and F-regions when first mentioned.

- "SQ": Define.

- Figure 4 show the equatorial electrojet estimated from the North-South component of the geomagnetic field (ΔH). Is H is horizontal component of the magnetic field? Needs to be defined. The authors should show the original H-field data, together with the variation in the N-S component of the field (ΔH). Also, it would be useful to remind readers of when the gamma-ray burst occurred, by over plotting it on this figure.

- The last paragraph of the paper is completely unrelated to the main focus of the work:

"Potentially, strong GRB events might abruptly deplete stratospheric ozone on a global scale³. Thomas et al. (2005)⁴ estimated that a global Ozone variation of 35% caused by a strong GRB or a similar cosmic event (statistically occurring at least once in a billion of years) would trigger widespread extinction. In fact, during the recovery phase of the Ozone (\sim 5 years) the Earth's surface would be exposed to up to a 3-fold increase in solar UVB flux, representing the most dangerous hazard for the biological system. Therefore, it is crucial to investigate how a strong-long lasting GRB impacts and modifies the Earth's atmosphere whose ionization stability plays a fundamental role for the evolution and endurance of human life¹."

Response to Reviewer #1

We thank the Reviewer for his/her comments and suggestions. In the revised version, all of Reviewer's suggestions have been considered.

- **GRB impacts on the upper ionosphere have not been observed to date, and it is not clear how GRBs could directly or indirectly produce disturbances in this region. The authors of this work report variations in ionosphere electric fields observed by the CSES satellite which was in a low Earth orbit over the side of the Earth that was illuminated by a particularly large GRB, which occurred on October 9. This is an interesting and significant result, but their physical explanation for how the D-region and F-region are linked is not well described and is not convincing.**

We thank the reviewer for his/her comment. To give a better and more quantitative explanation of the electric field variations observed in the top-side ionosphere as a consequence of the GRB occurrence, we developed a very simple analytical model ("toy-model") that, despite its simplicity is able to both represent our experimental measurements and give a physical explanation about the possible causes behind the E-field variation. It was inserted in the "methods" section. This kind of approach (analytical) let us understand that the E-field variation was caused by a change in the ionospheric conductivity induced by the strong photo-ionization driven by the strong October 9, 2022 GRB. For the sake of simplicity, in this work, we modelled the production rate induced by a GRB as a Gaussian impulsive function and we used a formalism directly related to the ratio between ion production rate for photo-ionization (α) and ionospheric loss rate for absorption (β). This allows the model to be independent (for the present analysis) of the calculation of a realistic photon production rate caused by a GRB, whose evaluation would require a Montecarlo approach and the estimation of the real top-side ionospheric ion cross-section, which is out of the scope of the present work. A more accurate modelling of the effect of a GRB on the top-side ionospheric electric field is in progress and will be presented in a forthcoming paper. Anyway, as you can see from the new figure inserted in the paper (that we reported below, too), for $\alpha/\beta < 3$ the effect of a GRB is negligible. To obtain results similar to what happens on October 09, 2022, our model requires a production-loss ratio greater than 5.

So, the analytical ionospheric electric field model we developed confirmed both our results and the hypothesis that the interaction between GRB and top-side ionosphere is a threshold process [Inan et al., 2007; Fishman and Inan, 1988; Mondal et al., 2010]. In addition, our model results suggest that

such a threshold depends strictly on the ratio between the ion production by photo-ionization process and ionospheric loss rate, but also on the time duration of the process itself. On the basis of this, we have changed the discussion and the conclusion of our paper. Of course, the experimental result related to the variation of the E-field is unchanged.

- **In addition, the authors describe total electron content (TEC) and estimate the equatorial electrojet, but it is not clear from these data (Figure 3 and 4) if they effects are significant and therefore if they bare any connection to the SID created by the GRB in the D-region.**

We changed the colour palette of figure 3 to highlight the TEC increase recorded on October 9 compared to the day before and after at the same time. Concerning the EEJ estimation, we have upgraded Figure 4 with the addition of a new box (i.e. Box A) in which we show the original H-field data. We also added a vertical line to highlight the arrival time of the GRB. Such a choice allows to better understand the causal link between the GRB occurrence and the EEJ variation. In fact, as we now stated in the new version of the manuscript, Figure 4B shows the comparison between the equatorial electrojet, estimated in terms of the variation of the North-South component of the geomagnetic field (H), calculated for a solar quiet day (October 12, 2022, black line) and for the day of the GRB occurrence (October 9, 2022, red line). It can be seen that the occurrence of the GRB221009A generated a perturbation of the EEJ. Indeed, superimposed to the long term variation, featured in both days and characterized by a minimum around both dawn and dusk, and by a maximum around the noon, at ~13:21 UT a low frequency (~0.35 mHz) fluctuation appears. Such a variation is more clear in the original magnetometer data used for the EEJ evaluation (Figure 4A). In fact, looking at Tatuoka data (box A right panels) which is located inside the EEJ, we can see that during quiet conditions (right-upper panel) the geomagnetic field reaches its maximum values around the local noon remaining almost stable for ~2.5 hours before decreasing down as the station approaches the local dusk [Chapman and Rao, 1965]. Differently, on October 9, before the GRB occurrence, as expected the H field reaches its maximum value, but, around 13:21 UT, almost in coincidence with the GRB occurrence, instead of remaining stable, it starts to fluctuate with a low frequency of ~0.35 mHz. This is a strong indication of the direct connection between the GRB and the EEJ variation.

Minor concern:

- **The paper does not flow well and is challenging to read due to its poor structuring. I would encourage the authors to better organise the paper so that it is easier of the reader to appreciate their observations and results. While the results are of potential interest, the data are not well presented and the physical link between the D- and F-regions is not convincing.**

We thank the reviewer for his/her comments. We completely revised the manuscript adding a new paragraph about modelling ionospheric electric field changes induced by a GRB which was able to give a first quantitative physical explanation of the experimental observations we made at 500 km. In addition, we added two new figures: one for the model result and one for the Equatorial electrojet estimation. Finally, we completely revised the discussions and the conclusions trying to let our manuscript to be more readable.